# A stable zirconium based metal-organic framework for specific recognition of representative polychlorinated dibenzo-*p*-dioxin molecules

Bin Wang [1], Peilong Wang [2], Lin-Hua Xie [1], Rui-Biao Lin [3], Jie Lv [1], Jian-Rong Li [1] & Banglin Chen [3]

Polychlorinated dibenzo-*p*-dioxins (PCDDs), as a class of persistent and highly toxic organic pollutants, have been posing a great threat to human health and the environment. The sensing of these compounds is important but challenging. Here, we report a highly stable zirconium-based metal-organic framework (MOF), $Zr_6O_4(OH)_8(HCOO)_2(CPTTA)_2$ (BUT-17) with one-dimensional hexagonal channels and phenyl-rich pore surfaces for the recognition and sensing of two representative PCDDs, 2,3-dichlorodibenzo-*p*-dioxin (BCDD) and 2,3,7,8-tetrachlorodibenzo-*p*-dioxin (TCDD), based on the fluorescence quenching. BUT-17 exhibits high sensing ability with the detection limits as low as 27 and 57 part per billion toward BCDD and TCDD, respectively, and is very selective as well without the interference of similar compounds. The recognition of BUT-17 toward BCDD is demonstrated by single-crystal structure of its guest-loaded phase, in which the fluorescence-quenched complexes form between the adsorbed BCDD molecules and the MOF host through $\pi$-$\pi$ stacking and hydrogen bonding interactions.

[1] Beijing Key Laboratory for Green Catalysis and Separation and Department of Chemistry and Chemical Engineering, College of Environmental and Energy Engineering, Beijing University of Technology, 100124 Beijing, P. R. China. [2] Institute of Quality Standards and Testing Technology for Agro-products, Chinese Academy of Agricultural Sciences, 100081 Beijing, P. R. China. [3] Department of Chemistry, University of Texas at San Antonio, One UTSA Circle, San Antonio, TX 78249-0698, USA. Correspondence and requests for materials should be addressed to J.-R.L. (email: jrli@bjut.edu.cnan) or to B.C. (email: banglin.chen@utsa.edu)

Molecular recognition is a fundamental phenomenon in biological and chemical systems and has played a very important role in the biological and chemical transformations. Extensive researches on supramolecular molecules and framework materials, particularly on metal-organic framework (MOF) materials, have revealed that we can readily make use of the molecular recognition for a variety of applications, ranging from gas storage[1–3], gas separation[4–7], catalysis[8–11], sensing[12–14], drug delivery[15,16], and photonic materials[17,18] to the structure determination of some very important chiral molecules[19]. In fact, quite a few open metal sites and specific functional sites on the pore surfaces have been developed for their molecular recognition and thus different applications. For example, the iron(II) sites on Fe-MOF-74 (Fe$_2$(dobdc), bobdc$^{2-}$ = 2,5-dioxido-1,4-benzenedicarboxylate) have been utilized to bind more ethylene than ethane[20]; while the peroxide iron(III) sites on Fe$_2$(O$_2$)(dobdc) have been realized to bind more ethane than ethylene, for their very important ethylene/ethane separation[21]. Recent studies have shown that the transformation of one specific site to another one (Lewis acid site) can lead to a remarkable CO$_2$ catalytic fixing, opening a new avenue of developing CO$_2$ chemical fixing catalysts[22]. Other than general functional sites to direct their interactions with guest substrates, pore confinement has also been utilized to capture specific substrates and thus to develop functional materials, as demonstrated in our MOF materials for the polarized three-photon pumped laser[23]. MOFs now have been emerging as a very powerful platform to introduce molecular recognition and thus to develop functional materials.

Polychlorinated dibenzo-$p$-dioxins (PCDDs), as a class of persistent and highly toxic organic pollutants (POPs), have been posing a great threat to human health associated with chloracne, immune suppression, nervous system function disorder, and abnormal endocrine[24,25]. PCDDs are mainly formed as unintended by-products of anthropogenic activities such as waste incineration, industrial production of herbicides and wood preservatives, and historically in chloralkali processes[26,27]. Due to significant improvements in emission controls and strict regulations, the amount of PCDDs released into the environment largely reduced in recent years. However, because of their structural resistance to microbial metabolism, environmental transformation processes, and low water solubility, PCDDs were still dispersed in soils and sediments at relatively high level[28–30]. Figure 1 shows the two representative PCDDs (BCDD and TCDD), which are composed of two phenyl groups bridged by oxygen atoms, and the H atoms on the phenyl groups are replaced by Cl atoms. Their hydrogen bonding and/or $\pi$–$\pi$ stacking interactions with the host porous materials might be not as strong as the typical solvent molecules, such as water, methanol, $N$,$N'$-dimethylformamide (DMF), and benzene. Therefore, it will be very challenging to realize the specific recognition of PCDDs in a suitable porous MOF. Given the fact that Zr-MOFs are very stable, we screened several known Zr-MOFs with different structures/

topologies and porosities for such a purpose[31]. We realized that those MOFs with isolated cages can barely recognize and thus sense one of the two representative PCDD molecules, BCDD, apparently attributed to their mismatch of the pore spaces and the absence of the $\pi$–$\pi$ stacking interactions with the BCDD molecules. Among these examined MOFs, the NU-1000[32] of the **csq-a** topology, which shows certain degree of recognition and sensing is really encouraging. This MOF is highly porous, so the pore confinement and/or pore match should not be so efficient to capture BCDD. From its structure, we speculated that the 1D channels for the straightforward access of the guest BCDD molecules and the pore surfaces composed of pyrene molecules for the $\pi$–$\pi$ stacking interactions with the BCDD molecules might play the important role for the recognition of BCDD. We thus focused on those Zr-MOFs of the **csq-a** topologies with open 1D channels which are large enough to encapsulate BCDD molecules and with pore surfaces of rich aromatic rings to induce their $\pi$–$\pi$ stacking interactions with the BCDD molecules.

In this work, we chose a smaller organic linker H$_4$CPTTA (5′-(4-carboxyphenyl)-[1,1′:3′,1″-terphenyl]-3,4″,5-tricarboxylic acid) to construct the iso-reticular Zr-MOF Zr$_6$O$_4$(OH)$_8$(H-COO)$_2$(CPTTA)$_2$ (BUT-17, BUT = Beijing University of Technology) of the **csq-a** topology and to narrow down the pore sizes of the 1D channels for better pore confinement. Detailed studies show that BUT-17 can indeed recognize BCDD molecules very well. In addition, BUT-17 also shows notable recognition ability toward TCDD, the most toxic PCDD. BUT-17 also exhibits very selective and sensitive luminescent sensing for BCDD molecules. Its detection limits toward BCDD and TCDD are very low down to 27 and 57 ppb, respectively. Single crystal X-ray analysis and density functional theory (DFT) calculations demonstrate that the open 1D channels with suitable pore sizes for the pore confinement, the aromatic pore surfaces for their $\pi$–$\pi$ stacking with the guest molecules and hydrogen bonding interactions collaboratively enforce the host BUT-17 for its specific recognition and detection of BCDD and TCDD molecules.

## Results

**Material screening and the design of BUT-17.** For the recognition of PCDDs, Zr-MOFs are promising candidates because of the following reasons: (1) generally, Zr-MOFs possessing good chemical stability and thus are suitable for the recognition and sensing of environmental pollutants under harsh conditions; (2) the fluorescence of Zr-MOFs is ligand-centred, and this single fluorescent source makes Zr-MOFs easily to be tailored for optimizing sensing performance; (3) since most Zr-MOFs are constructed from Zr$_6$ carboxylate clusters, the topologic diversity of them is limited to some extent[31]. Through the rational design of the ligands and the connectivity of Zr$_6$ clusters, the Zr-MOFs with expected structure can be constructed. In this work, firstly, we checked the sensing abilities of several Zr-MOFs with different framework structures and pore sizes including UiO-66[33], BUT-39[34], BUT-12[35], BUT-66[36], BUT-15[37], NU-1000[32], and NU-1003[38] toward BCDD (Supplementary Figs. 1–7). Fluorescence-quenching titrations were performed through the piece by piece addition of BCDD solution in hexane to hexane where the selected Zr-MOF was dispersed. As shown in Supplementary Figs. 15–21, the addition of BCDD has little effect on the fluorescence of BUT-66 and NU-1003, but it quenched the fluorescence of UiO-66, BUT-39, BUT-12, and BUT-15 in a small degree. Best quenching efficiency of 25% was observed in NU-1000. In our previous study, it was found that the pores of MOFs can play an important role in the enrichment of analytes, thus enhancing the sensing ability toward the analytes[35]. For the selected Zr-MOFs with isolated cages such as UiO-66 and BUT-39, the surfaces of

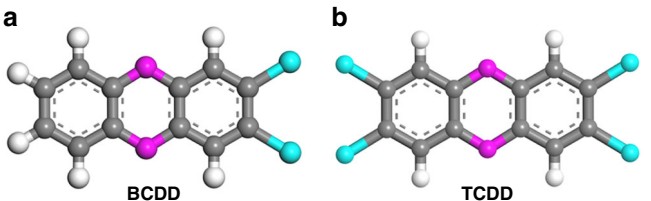

**Fig. 1** Two representative PCDD molecules. **a** 2,3-dichlorodibenzo-$p$-dioxin (BCDD) and **b** 2,3,7,8-tetrachlorodibenzo-$p$-dioxin (TCDD). Color code: Cl turquoise, C gray, O magenta, and H white

cages are covered by organic ligands, which prevent the access of BCDD molecules. Even though BUT-12 contains relatively large cages, which can allow the entrance of BCDD molecules, the structural mismatch of the BCDD molecules and the ligands in BUT-12 might lead to inefficient interactions, thereby its poor sensing ability toward BCDD. Interesting results were observed in NU-1003 and -1000. The two MOFs are isostructural with a **csq-a** topology and the only difference between them is that the former has larger hexagonal channel size (44 versus 35 Å, Supplementary Figs. 6 and 7). The hexagonal channels of the two MOFs are large enough for BCDD molecules entering and the highly conjugated fluorescent pyrene-based ligands in the two MOFs could well-match and form $\pi$–$\pi$ stacking interactions with the conjugated BCDD molecules. However, their sensing abilities toward BCDD are different. As mentioned above, the addition of BCDD has almost no effect on the fluorescence of NU-1003, and the quenching efficiency of 25% was observed in NU-1000. This phenomenon might be due to too large pores in the highly porous NU-1003 and -1000 to achieve an efficient pore confinement, which thus leads to relative weak interactions with BCDD molecules. As mentioned above, the channel size of NU-1003 is larger than that of NU-1000, thus, the interactions between BCDD molecules and NU-1003 should be weaker than that of NU-1000, thereby leading to worse sensing ability of the former. Further decreasing the channel size of NU-1000 might lead to stronger interactions and enhanced sensing performance. After checking the literatures, we found that previously reported Zr-MOFs[31,39–41] with the same topology as NU-1000; however, having equal or even larger pores than that of NU-1000, might not be so good for the sensing application of BCDD.

To further narrow down the pore sizes of the NU-1000, we designed a smaller ligand, 5′-(4-carboxyphenyl)-[1,1′:3′,1′′-ter-phenyl]-3,4′′,5-tricarboxylic acid (H$_4$CPTTA, see Fig. 2a and Supplementary Fig. 22). With this ligand, a new Zr-MOF, [Zr$_6$O$_4$(OH)$_8$(HCOO)$_2$(CPTTA)$_2$] (BUT-17) with expected **csq-a** topology was constructed, which contains smaller pores than those in NU-1000 (Fig. 2c, d). Stick-shaped single crystals of BUT-17 were obtained through the solvothermal reaction of H$_4$CPTTA and ZrCl$_4$ in the presence of formic acid as competing

reagent in DMF. Single-crystal X-ray diffraction shows that BUT-17 is isostructural to NU-1000 and -1003, but with shrunken pore diameter (Fig. 2c and Supplementary Table 1). The diameter of the hexagonal pore in BUT-17 is 24 Å, smaller than those of NU-1000 (35 Å) and NU-1003 (44 Å) (Supplementary Fig. 23). The total solvent-accessible volumes in BUT-17 framework was estimated to be 72.5% of its unit-cell volume[42]. Thermogravi-metric analysis (TGA) curves show that BUT-17 is stable up to *ca.* 410 °C (Supplementary Fig. 24). The permanent porosity of BUT-17 was then confirmed by N$_2$ adsorption at 77 K (Supplementary Fig. 26). Saturated N$_2$ uptake of 657 cm$^3$ g$^{-1}$ was achieved, and the evaluated Brunauer-Emmett-Teller (BET) surface area was 1790 m$^2$ g$^{-1}$ (Supplementary Figs. 27–29). The experimental total pore volume of BUT-17 was 1.02 cm$^3$ g$^{-1}$, being in close to the calculated value of 1.03 cm$^3$ g$^{-1}$[42]. Based on the N$_2$ adsorption data, the pore size distribution was calculated by DFT method, which gave two types of pores of 9 and 24 Å (Supplementary Fig. 26, inset), being consistent with the observation from crystal structure. In addition, BUT-17 displays good chemical stability in water, concentrated HCl aqueous solution, NaOH aqueous solution (pH = 12) at room temperature, as well as in boiling water, as confirmed by the powder X-ray diffraction (PXRD) measurements (Supplementary Fig. 25). Furthermore, the N$_2$ sorption isotherms for the samples after being soaked in water, boiling water, pH = 1 HCl aqueous solution, and pH = 10 NaOH aqueous solution for 24 h were measured and almost the same isotherms were observed as that of the pristine sample, further demonstrating the good chemical stability of BUT-17 under these conditions (Supplementary Fig. 26).

**Sensing of BCDD and TCDD.** Then, we explored the sensing ability of BUT-17 toward BCDD. As shown in Fig. 3 and Supplementary Fig. 30, the fluorescence-quenching efficiency of BUT-17 toward BCDD is 87%, greatly higher than that of NU-1000 (25%) under the same conditions. In addition, we also tested the sensing ability of BUT-17 toward TCDD, the most toxic PCDDs. Under the same conditions, TCDD also give high quenching efficiency of 73% for BUT-17 (Fig. 3 and Supple-mentary Fig. 31). The fluorescence-quenching efficiency can be

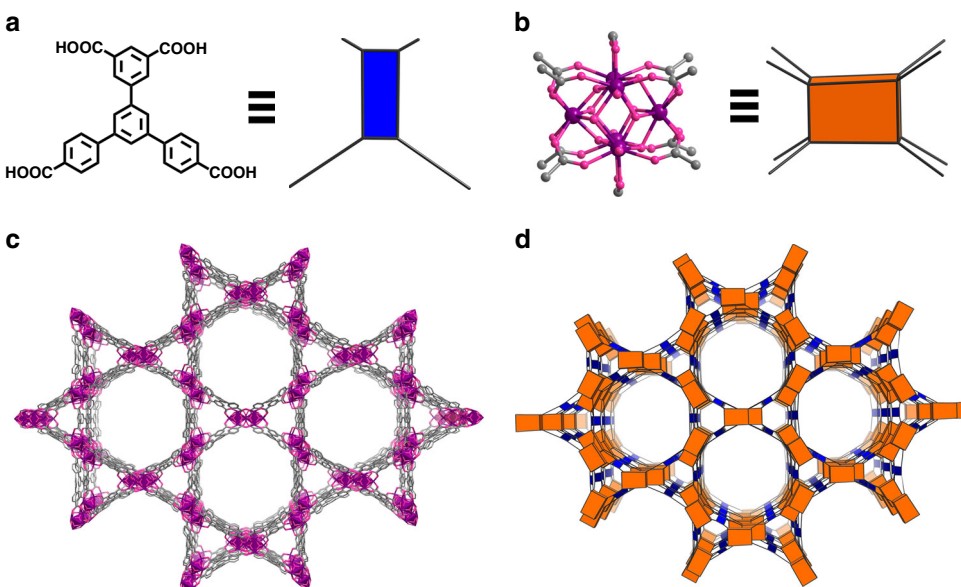

**Fig. 2** Reticular construction and the structure of BUT-17. **a** CPTTA$^{4-}$ ligand and its simplified 4-connected node, **b** Zr$_6$O$_4$(OH)$_8$(HCOO)$_2$(COO)$_8$ cluster and its simplified 8-connected node, **c** framework structure, and **d** the augmented *sqc* net. Color code: Zr violet, C gray, and O magenta, H atoms are omitted for clarity

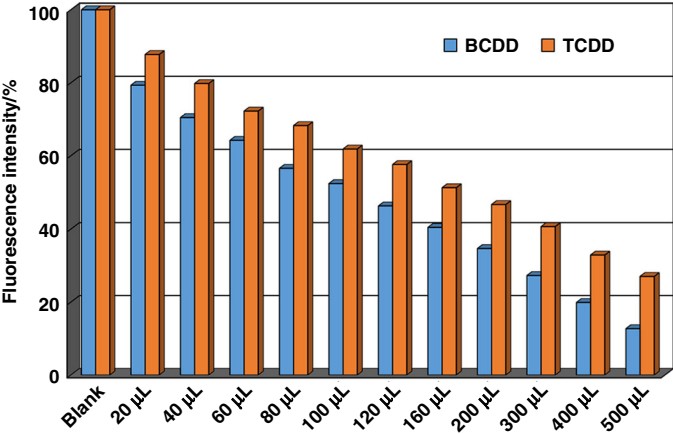

**Fig. 3** Fluorescence-quenching titrations of BUT-17 by TCDD and BCDD. Changes of the fluorescence of BUT-17 dispersed in hexane upon incremental addition of TCDD and BCDD (100 ppm)

quantitatively evaluated by the Stern-Volmer (SV) equation[43]: $(I_0/I) = K_{sv}[Q] + 1$, in which $I_0$ and $I$ are the luminescence intensities before and after addition of the analyte, respectively, $K_{sv}$ is the quenching constant ($M^{-1}$), and $[Q]$ is the molar concentration of the analyte. As indicated in Supplementary Figs. 32 and 33, the SV plots for BCDD and TCDD are nearly linear at low concentration ranges, but subsequently deviate from linearity and bend upwards at higher concentrations. In addition, at low concentrations, the decrease of the maximum fluorescence emission wavelength of BCDD and TCDD all show good linearity and the evaluated $K_{sv}$ values are as high as 37228 and 22251 $M^{-1}$ respectively (inset of Supplementary Figs. 32 and 33). Based on the $K_{sv}$ values and the standard deviations ($S_b$) from three repeated fluorescent measurements of blank solutions (Supplementary Fig. 34), the detection limits ($3S_b/K_{sv}$) of BUT-17 toward BCDD and TCDD in hexane were calculated to be 0.11 and 0.18 $\mu$M (corresponding to 27 and 57 ppb), respectively. It should be pointed that, till now, the detection of PCDDs is mainly based on analytical method such as high-resolution gas chromatography tandem high-resolution mass spectrometry (HRGC-HRMS) and bio-analytical methods such as biomarkers, bioassays, and enzyme immunoassays (EIAs)[44]. These methods are highly sensitive toward PCDDs and the detection limits toward PCDDs are low at part per trillion (ppt) level. However, they also suffer from some disadvantages. For instrumental analysis, it usually requires expensive equipment and highly specialized personnel and provides limited sample throughput; for bio-analytical methods, the degree of reliability (the relationship between chemical information and bioassay information) is a question. Therefore, the development of convenient, cost-effective PCDDs detection methods is of importance, and has significant impact on global environment protection and food safety. Fluorescence sensing based on the change in fluorescence readout induced by sensor-analyte interactions is a powerful detection method. To the best of our knowledge; however, there is no report regarding the detection of BCDD and TCDD using the fluorescence sensing method. Although the ppb-level detection limits of BUT-17 toward BCDD and TCDD are relative higher than those of instrumental methods (ppt-level) to some extent; however, we present here the first example of sensing of BCDD and TCDD based on the fluorescence sensing method.

In the practical detection of PCDDs, some organic molecules with similar structures such as polychlorinated biphenyl (PCBs) and poly brominated diphenyl ethers (PBDEs) are coexisted, and often affect the sensing result. Thus, the selective sensing of PCDDs from these organic molecules is important. In this work, several PCBs and PBDEs including 3,5-dichlorobiphenyl (PCB14), 3,3′,4,4′-tetrachlorobiphenyl (PCB 77), 1,2,3-trichloro-4-(3,4-dichlorophenyl)benzene (PCB105), 3,3′,4,4′,5,5′-hexachlorobiphenyl (PCB 169), 2,2′,3,5′-tetrachlorobiphenyl (PCB44), 2,3′,4,4′,5′-Pentachlorobiphenyl (PCB118), 4,4′-oxybis (1,2-dibromobenzene) (PBDE77), and 6,6′-oxybis(1,2,3,4,5-pentabromobenzene) (PBDE209) were selected as the interferents to evaluate their influences on sensing BCDD and TCDD by BUT-17 (Supplementary Fig. 35). It should be noted that, PCB14 and PCB77 has similar structure to BCDD and TCDD, respectively (Supplementary Fig. 35). The difference of them is the presence of a dioxane ring in the latter two molecules, which binding two adjacent benzene rings into a coplanar molecule (Fig. 1). PBDE47 also has similar structure to TCDD. As shown in Supplementary Figs. 36–43, the addition of PCB14, PCB118, PCB105, PCB169, and PBDE77 has little effect on the fluorescence of BUT-17, respectively. PCB77 and PCB44 can quench the fluorescence of BUT-17 to some extent, with the quenching efficiency of 10% and 14%, respectively, far smaller than that of BCDD and TCDD (being 87 and 73%, respectively) under the same conditions (Supplementary Fig. 44). The above results show that BUT-17 has high quenching efficiency toward BCDD and TCDD, but poor sensing abilities toward other interferents with similar structures. Motivated by these findings, we checked the sensing selectivity for BCDD in the presence these interferents. Firstly, equal amount of selected interferents (200 ppm for each analyte) were mixed to from a mixture. During the fluorescence titration experiments, the fluorescence spectra for BUT-17 dispersed in hexane were initially recorded. To this system, the above mixture of interferents was initially added and then followed by BCDD (100 ppm) and the corresponding emissions were monitored. As can see from Supplementary Fig. 45, the emission intensity of BUT-17 quenched 11% in the presence of the interferent mixture. Upon introducing BCDD to the mixture of the MOF and the interferent, the fluorescence of BUT-17 quenched 84%. This reveals that the interference from other analytes can be neglected, convincing the high selectivity of BUT-17 toward the BCDD.

The quenching process of BUT-17 toward BCDD and TCDD was further studied with transient fluorescence method (Supplementary Fig. 46). The emission decay of BUT-17 was fit best by a biexponential function with a shorter term ($\tau 1 = 7.472$ ns) and a longer term ($\tau 2 = 18.15$ ns, Supplementary Table 3). The biexponential decay could be explained by the inhomogeneous rotation or flipping kinetics of phenyl ring in BUT-17[45]. After

socked in the hexane solution of BCDD or TCDD, the lifetimes of the two terms of BUT-17, as well as their relative contributions, were almost unchanged (Supplementary Fig. 46 and Supplementary Table 3). The invariant fluorescence lifetime is typical for a static quenching pathway[46]. In the static quenching mechanism, the sensor would form a non-fluorescent complex with the analyte in the ground state, and this pre-association determines the quenching efficiency[47–49].

Since the great threaten of PCDDs to human health and fat-soluble nature of these compounds, even small concentration of PCDDs in water can lead to the accumulation of them in food chains. The removal of small amount of PCDDs in water system is quite important. We thus explored the removal ability of BUT-17 toward BCDD and TCDD in acetone aqueous solution (water: acetone = 1:1). As shown in Supplementary Fig. 47, the area of the peak representing BCDD in gas chromatography-mass spectrometer (GC-MS) decreased from 154492 to 41675, and that of TCDD decreased from 2018 to 1353, respectively, after adsorption for 2 h using BUT-17 as adsorbent at room temperature. Thus, the removing efficiencies of BUT-17 toward BCDD and TCDD were calculated to be 73% and 33%, respectively.

**The sensing mechanism exploration**. In order to better understand the fluorescence-quenching effect of BUT-17 toward BCDD and TCDD, the quenching mechanism was proposed. For comparison, the quenching mechanism of other Zr-MOFs was also proposed. Generally, the quenching on fluorescent MOFs by organic molecules are mainly explained by three mechanisms: fluorescence resonance energy transfer (FRET), photoinduced electron transfer (PET), and dynamic/static quenching mechanism[13]. FRET occurs only when the absorption spectrum of the analyte overlaps with the emission spectrum of the sensor. As shown in Supplementary Fig. 48, the spectral overlap between the absorption band of BCDD and TCDD and the emission spectra of BUT-17 as well as the selected Zr-MOFs is very limited, which hinders the energy transfer from MOFs to TCDD or BCDD molecules, indicating that FRET does not likely play a role in the TCDD or BCDD detection. In addition, we found that the emission spectrum of UiO-66 is closer to the adsorption spectra of TCDD or BCDD. For the other Zr-MOFs, the ligands are mainly based on large conjugated systems containing several benzene rings, and the conjugate degree of these MOFs is larger than that of UiO-66. Thus, the emission spectra of these Zr-MOFs will shift to the higher wavelengths[50]. Adjusting the fluorescence emission spectra of Zr-MOFs to overlap the adsorption band of BCDD is almost impossible. PET process works depends on the energy levels of the lowest unoccupied molecular orbitals (LUMOs) orbitals of the MOF and that of the analyte. If the LUMO orbital energy level of the MOF is higher than that of analyte, then the excited electron of the MOF could transfer to the LUMO orbital of the analyte, resulting in the fluorescence quenching. Thus, we calculated the orbital energies of the LUMOs and highest occupied molecular orbitals (HOMOs) of BCDD, TCDD, BUT-17, as well as the selected Zr-MOFs by DFT method using Material Studio software. Due to the fluorescence of Zr-MOFs is ligand centered, so the positions of their LUMOs and HOMOs could be represented respectively by the LUMO and HOMO of the ligands[31]. As shown in Supplementary Fig. 49 and Supplementary Table 4, the LUMO orbital energy levels of the TCDD and BCDD are higher than that of BUT-17 and those of the tested Zr-MOFs, indicating that the electron injection from excited MOF to the TCDD or BCDD molecule is thermodynamically forbidden. Therefore, to detect TCDD and BCDD by using fluorescent Zr-MOFs, the dynamic/static

quenching seems to be the only mechanism. Generally, dynamic quenching occurs mainly when there is a collision between a guest molecule and the heteroatoms in the fluorescent host, and in the static quenching mechanism, the sensor would form a non-fluorescent complex with the analyte in the ground state, and this pre-association determines the quenching efficiency[37,47]. Interestingly, TCDD and BCDD are highly conjugated planar molecules and contain rich heteroatoms and H atoms, which are easily form weak interactions such as π–π stacking or hydrogen bonding with some conjugated systems. No matter static or dynamic quenching process, however, the access of TCDD and BCDD molecules into the pores of MOFs is the prerequisite. As mentioned above, NU-1000 is promising for the recognition and sensing of BCDD; however, its large pores lead to relative weak interactions between the conjugated ligand and BCDD molecules. BUT-17 is isostructural with NU-1000, but the diameter of its hexagonal pores is smaller than that of NU-1000. The shrinking pore size could thus lead to the enhanced interactions between BCDD or TCDD molecules and BUT-17 framework, thus resulting in its better sensing ability.

To confirm this host-guest interactions, precise structure of BCDD included BUT-17 (BCDD@BUT-17) was determined by using single-crystal X-ray diffraction technique. Similar to the crystal sponge method developed by Fujita and co-workers[19], single crystals of guest-free phase of BUT-17 were firstly obtained by heating the single crystals of acetone-exchanged BUT-17 in an oven (150 °C) for 5 h. Subsequently, the guest-free single crystals of BUT-17 were immersed in the hexane solution of BCDD and allowed the solvent to evaporate slowly (Supplementary Fig. 50). When the solvent in the bottom is almost evaporated, a single crystal was taken out and placed on the single-crystal X-ray diffractometer and heated under the $N_2$ flow (50 °C) for 5 h to remove the residuary solvent in the pores of BUT-17. After that, a new set of diffraction data at 100 K was collected for BCDD@BUT-17. After structure determination, we found that in BCDD@BUT-17, the absorbed BCDD molecules can be well-identified and they are mainly located around the surface of the large hexagonal pores (Fig. 4a and Supplementary Table 2). The dihedral angle between the adsorbed BCDD molecule and the central coplanar of CPTTA$^{4-}$ ligand is 17°, and the distance of the central dioxane ring of BCDD and the benzene ring of the ligand is 3.3 Å, indicating the presence of the π–π stacking interactions (Fig. 4c and Supplementary Figs. 51 and 52). In addition, there exist hydrogen bonding interactions between the O atoms of the coordinated formic acid on the $Zr_6$ clusters and H atoms of BCDD molecules, the distance of which is 2.5 and 2.4 Å, respectively, (Fig. 4d and Supplementary Fig. 53). The O atoms of BCDD molecules also form hydrogen bonds with H atoms of the coordinated formic acid on $Zr_6$ clusters, the distance of which is 2.4 Å (Fig. 4d and Supplementary Fig. 53). Interestingly, the coordinated formic acids on the $Zr_6$ clusters play the role of fixing BCDD molecules. As shown in Fig. 4b and Supplementary Fig. 54, the position of the two formic acid molecules is a little higher than that of BCDD molecules which means that, like a "molecule clip", the two formic acid molecules can fix the BCDD molecules, further enhance the interactions between BCDD molecules and the ligands in BUT-17. The cooperation of π–π stacking, hydrogen bonding interactions, and the so-called molecules clip mentioned above thus lead to the relative strong affinity between BCDD molecules and the framework of BUT-17.

Considering that the formation of MOF-analyte complex will change the electron configuration of the whole system, we calculated the orbital energies of the HOMOs and LUMOs of BUT-17@BCDD and compared them with those of original BUT-17 by DFT (Supplementary Fig. 55). It was found that the HOMO orbitals of BUT-17 are mainly around the lower part of the

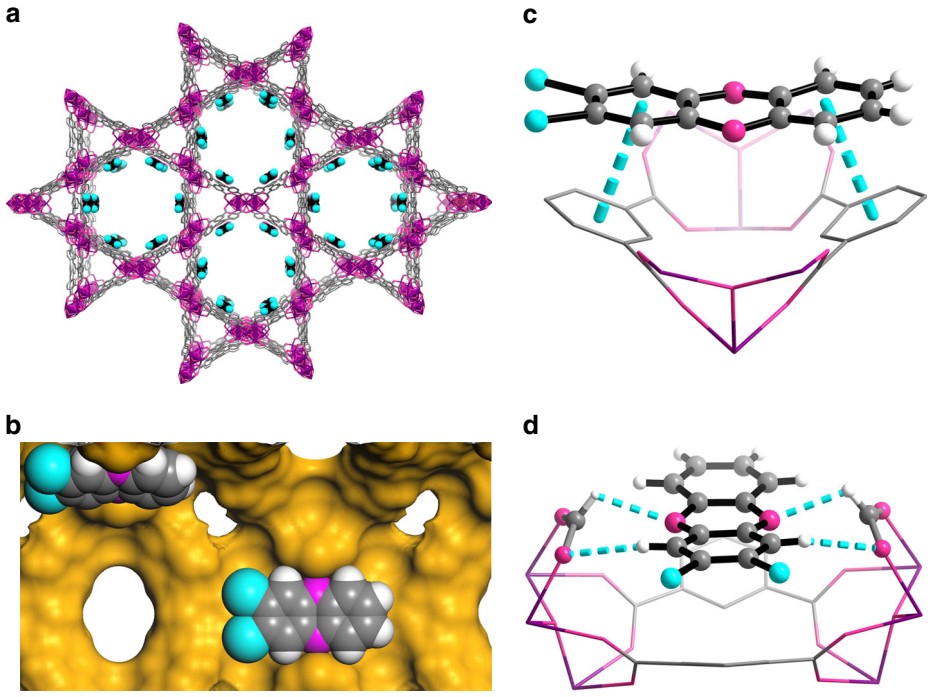

**Fig. 4** Crystal structure of BCDD loaded BUT-17. **a** BCDD adsorption sites in BUT-17; **b** Selected fragments highlighting the "molecule clip" in BCDD@BUT-17; Selected fragments highlighting the **c** $\pi$-$\pi$ stacking interactions; and **d** hydrogen bonding interactions in BCDD@BUT-17. Color code: Cl turquoise, Zr purple, C gray, O magenta, and H white

ladder-shaped CPTTA$^{4-}$ ligand, and the LUMO orbitals of BUT-17 are mainly around the upper part of the ladder-shaped CPTTA$^{4-}$ ligand (Supplementary Fig. 55a, b). Interestingly, when BCDD molecules access the hexagonal pores of BUT-17 and form the complex with the ligands, the orbital distributions of this MOF-analyte system were changed. The HOMO orbitals of BCDD-CPTTA$^{4-}$ are mainly around the BCDD molecules, whereas the LUMO orbitals locates at the CPTTA$^{4-}$ ligands (Supplementary Fig. 55c, d). As shown in Supplementary Fig. 56, the ultraviolet-visible (Uv-vis) spectrum of BCDD is similar with that of BUT-17, and the maximum adsorption wavelength of them are both around 300 nm. Thus, under the excitation wavelength of BUT-17, the electron on the ground state of BCDD-CPTTA$^{4-}$ can absorb energy from the light to form excited electron. However, when the excited electron back to ground state, the absorbed energy should be released in non-radiative path instead of light, thus the complex cannot emission light. Such non-fluorescent complex quenched the fluorescence of BUT-17, and with the increasing amount of BCDD, more and more non-fluorescent complexes formed, leading to the continuous quenching of the fluorescence of BUT-17.

In addition, the regeneration is an important issue for a sensor. Therefore, we also explored the sensing ability of regenerated BUT-17 toward BCDD and TCDD. It was found that the quenching efficiencies of generated BUT-17 toward BCDD and TCDD are basically unchanged up to six cycles, demonstrating its good recyclability and stability for the sensing application (Supplementary Fig. 57).

## Discussion
In summary, a new stable fluorescent Zr-MOF with **csq-a** topological structure, BUT-17 has been designed, synthesized, and used in the recognition of two representative PCDDs (BCDD and TCDD). The detection limits of BUT-17 for BCDD and TCDD are estimated to be 27 and 57 ppb, respectively. The main

mechanism of the fluorescent quenching is believed to be the formation of non-fluorescent complex between the PCDDs and the ligands in BUT-17 through $\pi$-$\pi$ stacking and hydrogen bonding interactions, as proved by single-crystal X-ray diffraction and DFT calculation. The resulting new MOFs are thus potentially useful for the PCDDs sensing applications, which opens a new direction for practical applications making use of multifunctional MOFs.

## Methods
**Synthesis of BUT-17**. ZrCl$_4$ (48 mg, 0.20 mmol), H$_4$CPTTA (29 mg, 0.06 mmol), and formic acid (2.8 mL) were ultrasonically dissolved in 8 mL of DMF in a 20 mL Pyrex vial and sealed. The vial was then heated at 120 °C for 48 h in an oven. After cooling to room temperature, the resulting colorless crystals were harvested by filtration and washed with DMF and acetone, and then dried in air (yield 32 mg).

**Synthesis of other selected Zr-MOFs**. The detail of synthesizing other selected Zr-MOFs can be found in Supplementary Methods. The characterizations of these selected Zr-MOFs can be found in Supplementary Figs. 8–14.

**Fluorescence measurements**. TCDD and BCDD are highly toxically chemicals and should be handled carefully and the measurements should be carried out in the professional laboratory!

The finely grounded powder sample of MOF (5 mg) was immersed in 30 mL of hexane and ultrasonicated for 30 min to form stable turbid suspension, then 1 mL of it was added to a cuvette. The fluorescence of MOF was measured in situ after incremental addition of freshly prepared analyte solutions (100 ppm). The mixed solution was stirred at constant rate during experiment to maintain its homogeneity. All the experiments were performed in triplicate, and consistent results are reported. The excitation wavelengths of BUT-17 and selected Zr-MOFs can be found in Supplementary Table 5.

In a selective sensing experiment, firstly, equal amount of interferents (200 ppm for each analyte) were mixed to form a mixture. During the fluorescence titration experiment, the fluorescence spectra for BUT-17 dispersed in hexane were initially recorded. To this system, the above mixture of interferents (500 μL) was initially added and then followed by TCDD or BCDD (100 ppm, 500 μL) and the corresponding emissions were monitored.

## Data availability
Crystallographic data for the structures reported in this Article have been deposited at the Cambridge Crystallographic Data Centre, under deposition numbers CCDC 1883541 (BUT-17) and 1883542 (BUT-17@BCDD). Copies of the data can be obtained free of charge via www.ccdc.cam.ac.uk/data_request/cif. All other data supporting the findings of this study are available within the article and its Supplementary Information, or from the corresponding author upon reasonable request.

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

## Acknowledgements
We acknowledge financial support from the National Natural Science Foundation of China (No. 51621003, 21576006, and 21771012).

## Author contributions
J.-R.L., B.C., and P.W. conceived and designed the experiment. B.W. conducted the synthesis and sensing measurements. L.-H.X. and R.-B.L. performed SXRD measurement and crystal structure analysis. J.L. performed PXRD and $N_2$ adsorption/desorption measurements. J.-R.L., B.C., and B.W. co-wrote the manuscript. All authors discussed the results and commented on the manuscript.

## Additional information

**Competing interests:** The authors declare no competing interests.

