## [Peer Review File · Nature Communications]

Reviewers' comments:

Reviewer #1 (Remarks to the Author):

In this manuscript, the authors reported a new Zr-MOF, namely BUT-17, and demonstrated that it can be used for the detection of toxic compounds BCDD and TCDD based on fluorescence quenching. The detection limits could reach ppb-level, and single crystal structures of MOF encapsulated with guest molecules were presented to support the proposed sensing mechanism. The presented sensing performance seems good, and the single crystal structure analysis is impressive. However, the MOF structure is isostructural to NU-1000, so the novelty is low from the materials point of view. It might be further improved and here are the detailed comments.

1. Chemical sensing using fluorescent MOFs based on fluorescence quenching has been well studied. The authors have listed several possible mechanisms and used single crystal structures and DFT calculations to support their claimed mechanism. However, those evidences are indirect. Advanced photophysical characterizations such as ultrafast spectroscopy should be conducted to provide solid evidence on any possible electron or energy transfer processes.
2. The authors focused on PCDD molecules as the analyte in this study. Although ppb-level sensitivity has been claimed, state-of-the-art sensing technologies for those molecules should be presented and compared to give the readers a better understanding of the sensing performance. In addition, the cited story of Ukrainian present seems unsuitable for scientific publication.
3. The sensing performance difference between BUT-17 and NU-1000 was attributed to the pore size, which is not very convincing. Compare experiments should be provided (e.g., MOFs with similar porous structures but different hexagonal channel sizes) to strengthen this argument.
4. The resolution and quality of the figures in the supporting information need to be improved.

Reviewer #2 (Remarks to the Author):

In this entitled work "A Highly Stable Zr-Metal-Organic Framework with Suitable 1D Hexagonal Channels and Phenyl Pore Surfaces for Specific Recognition and Detection of Two Representative Polychlorinated Dibenzop-dioxin (PCDD) Molecules", authors have demonstrated detection of two highly toxic chlorinated compounds (TCDD and BCDD) with a Zr-based metal-organic framework. Zr-MOF synthesized here namely BUT-17, has shown good chemical stability even in presence of acid and base medium. Current MOF was designed for the targeted application and shows good potential for the sensing of such PCDD over other well-known Zr-MOFs. Authors have provided reasonable mechanism for the selective sensing and most importantly supported by the crystal structure of the guest included MOF. After going through the manuscript, I feel this work is suitable for the publication in Nature Communications after minor revision.

The following issues should be addressed by the authors before final acceptance,

1. Although the solubility of both BCDD and TCDD in water is very low, but even small concentration of such compounds in water can lead to the accumulation of its in food chains. In this regard, authors should carry out another set of experiments where the removal performance of BUT-17 should be investigated in aqueous medium for each PCDDs.
2. Further, one "A-level error" is present in the check cif file of the BCDD encapsulated BUT-17. Either authors should try to remove this error or they include an explanation regarding the error in the cif file.
3. Relevant references, like other toxic organic pollutant sensing should be included in the manuscript e.g. J. Am. Chem. Soc. 2016, 138, 6204–6216, Angew. Chem. Int. Ed. 2013, 52 (10), 2881-2885, Chem. Soc. Rev. 2017, 46, 3242-3285.
4. Authors should include discussion regarding the general sources and use of PCDDs in the introduction section.

Response to Comments

Reviewer #1

General comments: In this manuscript, the authors reported a new Zr-MOF, namely BUT-17, and demonstrated that it can be used for the detection of toxic compounds BCDD and TCDD based on fluorescence quenching. The detection limits could reach ppb-level, and single crystal structures of MOF encapsulated with guest molecules were presented to support the proposed sensing mechanism. The presented sensing performance seems good, and the single crystal structure analysis is impressive. However, the MOF structure is isostructural to NU-1000, so the novelty is low from the materials point of view. It might be further improved and here are the detailed comments.

Response: Many thanks for these constructive comments and useful suggestions. According to the suggestions, we have carried out additional experiments, added related discussions, and carefully revised the manuscript. Point-by-point responses to the comments and suggestions are listed below.

Specific comment 1: Chemical sensing using fluorescent MOFs based on fluorescence quenching has been well studied. The authors have listed several possible mechanisms and used single crystal structures and DFT calculations to support their claimed mechanism. However, those evidences are indirect. Advanced photophysical characterizations such as ultrafast spectroscopy should be conducted to provide solid evidence on any possible electron or energy transfer processes.

Response 1: Thank you for the comment. Following your suggestion, we carried out the ultrafast transient absorption spectroscopy measurement (TA) using the pristine MOF solution and the BCDD containing MOF solution, respectively. With the help of two different groups, respectively who are professional in the ultrafast transient absorption study (Xinfeng Liu's group at National Center for Nanoscience and Technology and Cheng Wang's group at Xiamen University), we were able to do the measurement.

In order to prove if there is any electron transfer or energy transfer between BUT-17 (MOF) and BCDD, we would like to selectively excite BUT-17 and see if we can observe

the quenching of the excited state of BUT-17 by BCDD, together with any transient absorption peaks ascribed to the excited BCDD or as-obtained BCDD radical anion (BCDD⁻). As shown in Figure I, BUT-17 and BCDD have very similar absorption maximum at around 310 nm. Additionally, BUT-17 has a broader absorption peak shoulder extending to longer wavelength. To avoid the excitation of BCDD, the excitation wavelength we used were 345 and 360 nm. Unfortunately, after many attempts by the two groups mentioned above, we cannot detect any TA signals of BUT-17 using 345 and 360 nm excitation. As shown in Figure IIa and b, the signals should be the solvent response. However, when the excitation wavelength was changed to 300 nm, the absorption maximum of BUT-17, we can clearly see the transient absorption of BUT-17. As shown in Figure IIc, the peaks around 450 and 600 nm can be assigned to BUT-17. We assumed that BUT-17 cannot be excited by 345 or 360 nm light, which seems inconsistent with the UV-vis measurement. However, since BUT-17 is solid sample suspended in hexane, the scattering light should be considered. BUT-17 is a Zr-based MOF, and Zr₆ cluster usually don't contribute to the absorption. Thus, the absorption of BUT-17 is mainly based on the CPTTA⁴⁻ ligand. As shown in Figure I, no absorption peaks were observed for the ligand at the wavelength longer than 340 nm, which confirms our assumption that the absorption of BUT-17 at longer than 340 nm is due to the light scattering. In this case, it is impossible to only excite BUT-17 without exciting BCDD. Thus, this system seems not suitable for the femtosecond-picosecond transient absorption measurement.

Fig. I UV-vis spectra of BUT-17, BCDD, and H₄CPTTA in hexane.

Fig. II Ultra-fast transient absorption maps of BUT-17 (left) and corresponding kinetic traces at selected wavelengths (right) pumped at: (a) 345 nm; (b) 360 nm; and (c) 300 nm.

Specific comment 2: The authors focused on PCDD molecules as the analyte in this study. Although ppb-level sensitivity has been claimed, state-of-the-art sensing technologies for those molecules should be presented and compared to give the readers a better understanding of the sensing performance. In addition, the cited story of Ukrainian present seems unsuitable for scientific publication.

Response 2: Thanks for your valuable comment and useful suggestion.

(1) The detection and removal of PCDDs from the environmental and foodstuff samples are worldwide problems. Now, the detection of PCDDs is mainly based on instrument-analytical methods such as high-resolution gas chromatography tandem high-resolution mass spectrometry (HRGC-HRMS) and bio-analytical methods such as biomarkers, bioassays, and enzyme immunoassays (EIAs) (RSC Adv., 2016, 6, 55415). These methods are highly sensitive toward PCDDs and the detection limits of them are

low at ppt level, better than that of BUT-17 (ppb level). However, these methods also suffer from some advantages. For instrumental analysis, it requires expensive equipments and highly specialized personnel, and provides limited sample throughput; for bio-analytical methods, the degree of reliability (the relationship between chemical information and bioassay information) is a question. Therefore, the development of convenient, cost-effective PCDDs detection methods is of importance, and has significant impact on global environment protection and food safety. Fluorescence sensing based on the change in fluorescence readout induced by sensor-analyte interactions is a powerful detection method. To the best of our knowledge, however, there is no report regarding the detection of BCDD and TCDD using the fluorescence sensing method. Although the ppb-level detection limits of BUT-17 toward BCDD and TCDD are not comparable to those of instrumental methods (ppt-level) to some extent, however, we present here the first example of sensing of BCDD and TCDD based on fluorescence sensing method, opening a new way of developing detection method of PCDDs. We have added some related discussions in the revised manuscript, please see paragraph 1, page 8. The added paragraph is as following:

“It should be pointed that, till now, the detection of PCDDs is mainly based on instrument-based analytical methods such as high-resolution gas chromatography tandem high-resolution mass spectrometry (HRGC-HRMS) and bio-analytical methods such as biomarkers, bioassays, and enzyme immunoassays (EIAs)⁴⁴. These methods are highly sensitive toward PCDDs and the detection limits toward PCDDs are low at ppt level. However, they also suffer from some advantages. For instrumental analysis, it usually requires expensive equipment and highly specialized personnel and provides limited sample throughput; for bio-analytical methods, the degree of reliability (the relationship between chemical information and bioassay information) is a question. Therefore, the development of convenient, cost-effective PCDDs detection methods is of importance, and has significant impact on global environment protection and food safety. Fluorescence sensing based on the change in fluorescence readout induced by sensor-analyte interactions is a powerful detection method. To the best of our knowledge, however, there

is no report regarding the detection of BCDD and TCDD using the fluorescence sensing method. Although the ppb-level detection limit of BUT-17 toward BCDD and TCDD is relative higher than those of instrumental methods (ppt-level) to some extent, however, we present here the first example of sensing of BCDD and TCDD based on fluorescence sensing method.”

(2) The story of Ukrainian president has been removed from the revised manuscript. Thank you so much for your reminding!

Specific comment 3: The sensing performance difference between BUT-17 and NU-1000 was attributed to the pore size, which is not very convincing. Compare experiments should be provided (e.g., MOFs with similar porous structures but different hexagonal channel sizes) to strengthen this argument.

Response 3: Thanks for your comment and useful suggestion. After screening literatures, we found that among reported Zr-MOFs featuring *csq-a* topology, the hexagonal channel size of NU-1000 is the smallest. Thus, another isorecticular Zr-MOF, NU-1003 with the hexagonal channels larger than that of NU-1000 (44 versus 35 Å) was selected for the comparison (Figure III). PXRD measurement and N₂ uptake at 77 K both confirmed the successfully synthesis of NU-1003 (Figure IV). Then, this MOF was used for the sensing of BCDD following the same process as other selected Zr-MOFs. As shown in Figure V, the addition of BCDD almost has no effect on the fluorescence of NU-1003, which is in contrast with that of NU-1000 under the same conditions. The two MOFs have similar structure and the only difference of them is that the ligand used for the construction of NU-1003 is larger than that of NU-1000, which leading to a larger hexagonal channel size of the former. We thus assume that the “larger pore” of NU-1003 lead to weaker interactions between BCDD molecules and its framework. Thus, further decrease the channel size of NU-1000 should lead to stronger interactions and enhance the sensing performance. We have added these results and relevant discussion in the revised manuscript, please see paragraph 1 and page 6. The revised paragraph is as following:

“Interesting results were also found in NU-1003 and -1000. The two MOFs are isostructural with a *csq*-a topology and the only difference between them is that the former has larger hexagonal channel size (44 versus 35 Å, Supplementary Fig. 6 and 7). The hexagonal channels of the two MOFs are large enough for BCDD molecules entering and the highly conjugated fluorescent pyrene-based ligands in the two MOFs could well-match and form π - π stacking interactions with the conjugated BCDD molecules. However, the sensing abilities of them toward BCDD are different. As mentioned above, the addition of BCDD has almost no effect on the fluorescence of NU-1003, and the quenching efficiency of 25% was observed in NU-1000. This phenomenon might be due to too large pores in the highly porous NU-1003 and -1000 to achieve an efficient pore confinement, which thus leads to relative weak interactions with BCDD molecules. As mentioned above, the channel size of NU-1003 is larger than that of NU-1000, thus, the interactions between BCDD molecules and NU-1003 should be weaker than that of NU-1000, thereby leading to worse sensing ability of the former. Further decreasing the channel size of NU-1000 might lead to stronger interactions and enhanced sensing performance. After checking the literatures, we found that previously reported Zr-MOFs^{31,39-41} with the same topology as NU-1000, however, having equal or even larger pores than that of NU-1000, might not be so good for the sensing application of BCDD.”

Figure III. (a) Ligand for the construction of NU-1003, (b) 3D framework structure of NU-1003, and (c) hexagonal channel along *c* axis (Color code: C, black; O, red; and Zr, plum; H atoms on ligands are omitted for clarity).

Figure IV. (a) PXRD patterns of NU-1003 and (b) N_2 adsorption/desorption isotherms of NU-1003 at 77 K.

Figure V. The emission spectra of NU-1003 dispersed in hexane upon incremental addition hexane solutions of BCDD (100 ppm).

Specific comment 4: The resolution and quality of the figures in the supporting information need to be improved.

Response 4: Thank you for your suggestion. The resolution and quality of the figures in the SI have been improved.

Reviewer #2

General comments: In this entitled work “A Highly Stable Zr-Metal-Organic Framework with Suitable 1D Hexagonal Channels and Phenyl Pore Surfaces for Specific Recognition

and Detection of Two Representative Polychlorinated Dibenzop-dioxin (PCDD) Molecules”, authors have demonstrated detection of two highly toxic chlorinated compounds (TCDD and BCDD) with a Zr-based metal-organic framework. Zr-MOF synthesized here namely BUT-17, has shown good chemical stability even in presence of acid and base medium. Current MOF was designed for the targeted application and shows good potential for the sensing of such PCDD over other well-known Zr-MOFs. Authors have provided reasonable mechanism for the selective sensing and most importantly supported by the crystal structure of the guest included MOF. After going through the manuscript, I feel this work is suitable for the publication in Nature Communications after minor revision. The following issues should be addressed by the authors before final acceptance.

Response: Thanks for your positive comments and constructive suggestions. As you suggested, we have carried out additional experiments, added related discussions, and carefully revised the manuscript. Point-by-point responses to the comments and suggestions are listed below.

Specific comment 1: Although the solubility of both BCDD and TCDD in water is very low, but even small concentration of such compounds in water can lead to the accumulation of its in food chains. In this regard, authors should carry out another set of experiments where the removal performance of BUT-17 should be investigated in aqueous medium for each PCDDs.

Response 1: Thank you for your suggestion. According to the literature report, the solubility of BCDD and TCDD in pure water at 298 K is 14.9 and 0.2 $\mu\text{g L}^{-1}$ (corresponding to 14.9 and 0.2 ppt), respectively, which is very low (Environ. Sci. Technol. 1988, 22, 651). In this case, to obtain a TCDD aqueous solution, 2 mg TCDD should be added to 1×10^4 L pure water, which is difficult to operate and easily leads to big experimental error. In addition, the concentration of such a TCDD aqueous solution (0.2 ppt) is too low for any instrument to detect. Thus, to test the removal performance of BUT-17 toward BCDD and TCDD in water-containing system, we carried out the

additional adsorption experiments in water/acetone solutions. Experimentally, the aqueous solution (water: acetone = 1 : 1) of BCDD or TCDD with the concentration of 10 ppm was accurately prepared and its intensity and area were recorded using GC-MS. Then, 15 mg powder of BUT-17 was added in the above-mentioned aqueous solutions (10 mL for each). After adsorption for 2 h at 298 K, the powder of MOF was centrifuged and the intensity of residual BCDD or TCDD in the filtrate was recorded again using GC-MS. The removing efficiency was thus calculated based on the changes of the intensities before and after adsorption. As shown in Figure VI the intensity of BCDD decreased from 154492 to 41675, and that of TCDD decreased from 2018 to 1353, thus, the removing efficiency of BUT-17 toward BCDD and TCDD is 73 and 33%, respectively. We have added these results and relevant discussion in the revised manuscript, please see paragraph 2, page 11. The added paragraph is as following:

“Since the great threaten of PCDDs to human health and excellent fat solubility of these compounds, even small concentration of such compounds in water can lead to the accumulation of them in food chains. The removal of small amount of PCDDs in water system is quite important. We thus explored the removal ability of BUT-17 toward BCDD and TCDD in aqueous solution of acetone (water:acetone = 1:1). As shown in Supplementary Fig. 48, the intensity of BCDD in the acetone aqueous solution decreased from 154492 to 41675, and that of TCDD decreased from 2018 to 1353, respectively. Thus, the removing efficiencies of BUT-17 toward BCDD and TCDD were calculated to be 73, and 33%, respectively.”

Figure VI. The intensities of (a) BCDD before (left) and after (right) adsorption experiments, and (b) TCDD before (left) and after (right) adsorption experiments.

Specific comment 2: Further, one "A-level error" is present in the check cif file of the BCDD encapsulated BUT-17. Either authors should try to remove this error or they include an explanation regarding the error in the cif file.

Response 2: Thank you for your reminding. The structure of BUT-17@BCDD has been further refined and the "A-level error" has been removed, please check the checkcif report.

Specific comment 3: Relevant references, like other toxic organic pollutant sensing should be included in the manuscript e.g. J. Am. Chem. Soc. 2016, 138, 6204-6216, Angew. Chem. Int. Ed. 2013, 52 (10), 2881-2885, Chem. Soc. Rev. 2017, 46, 3242-3285.

Response 3: Thank you for your suggestion. The relevant references mentioned above have been added in the revised manuscript, see the ref. 35, 43, and 13, respectively. In addition, we further added another relevant reference (Acc. Chem. Res. 2017, 50, 2457-2469), see the ref. 14.

Specific comment 4: Authors should include discussion regarding the general sources and use of PCDDs in the introduction section.

Response 4: Thank you for your suggestion. PCDDs are mainly formed as unintended

by-products of anthropogenic activities including waste incineration, industrial production of herbicides and wood preservatives, and historically in chloralkali processes, and these compounds are useless for humans. The introduction for the general sources of PCDDs has been added in the revised manuscript. Please see paragraph 2, page 3. The added paragraph is as following:

“PCDDs are mainly formed as unintended by-products of anthropogenic activities including waste incineration, industrial production of herbicides and wood preservatives, and historically in chloralkali processes²⁶⁻²⁷. Due to significant improvements in emission controls and strict regulations, the amount of PCDDs released into the environment largely reduced in recent years. However, because of their structural resistance to microbial metabolism, environmentally transformation processes, low water solubility, PCDDs were dispersed in soils and sediments at levels of concern²⁸⁻³⁰.”

REVIEWERS' COMMENTS:

Reviewer #2 (Remarks to the Author):

In the revised version, authors have performed suggested experiment in water and got positive results and also included other suggested corrections. In my opinion, the manuscript is now suitable for publication in Nat Commun in its current form.

Response to Reviewers' Comments

Reviewer #2:

General comments: In the revised version, authors have performed suggested experiment in water and got positive results and also included other suggested corrections. In my opinion, the manuscript is now suitable for publication in Nat Commun in its current form.

Response: We highly appreciate the reviewer for the supportive comments.